

# Detection and phylogenetic analysis of adenoviruses occurring in a single anole species

Sofia R. Prado-Irwin[1,*], Martijn van de Schoot[1,2,*] and Anthony J. Geneva[1]

[1] Department of Organismic and Evolutionary Biology, Harvard University, Cambridge, MA, USA
[2] Plant Ecology and Nature Conservation Group, Wageningen University, Wageningen, Netherlands
* These authors contributed equally to this work.

## ABSTRACT

Adenoviruses (AdVs) infect a wide range of hosts, and they have undergone recent and ancient host transfers multiple times. In reptiles, AdVs have been found in many captive individuals, and have been implicated in morbidity and mortality in several species. Yet the pathogenicity, transmission, phylogenetic distribution, and source of AdVs in the environment are still unknown. We therefore chose to opportunistically sample deceased captive *Anolis sagrei* individuals that were collected from different populations in the Bahamas and the Cayman Islands, as well as fecal samples from one island population, to explore the disease dynamics and diversity of adenovirus infecting *A. sagrei* populations. We found that adenovirus infection was present in our captive colony at low prevalence (26%), and was likely not the primary cause of observed morbidity and mortality. Among the 10 individuals (out of 38 sampled) which tested positive for adenovirus, we identified four adenovirus clades, several of which are distantly related, despite the close relationships of the *A. sagrei* host populations. These results suggest that while adenovirus may not be highly prevalent in the wild, it is present at low levels across much of the range of *A. sagrei*. It may undergo frequent host switching across both deep and shallow host divergences.

# INTRODUCTION

Viruses have long been known to cause morbidity and mortality throughout eukaryotes, and while much of the field of virology has been focused on human health, recent work has also begun to explore viral infections in other taxa (*Davison, Wright & Harrach, 2000*; *Harrach, 2000*; *Benkő et al., 2002*; *Rivera et al., 2009*; *Kovács et al., 2010*). In reptiles, pet trade husbandry has allowed for the study of viruses infecting controlled captive populations, providing to a robust understanding of several diseases in captivity (*Raymond et al., 2003*; *Garner et al., 2008*; *Abbas et al., 2011*; *Hyndman & Shilton, 2011*; *Doneley, Buckle & Hulse, 2014*; *Bak, Yeonsook & Woo, 2018*). However, field-based studies of these pathogens have lagged. With novel emerging

Corresponding author
Sofia R. Prado-Irwin,
spradoirwin@gmail.com

infectious diseases causing population declines in many taxa (*Ariel, 2011*; *Marschang, 2011*; *Marschang, Stohr & Allender, 2016*; *Lorch et al., 2016*), it is important to understand the prevalence and effect of viral pathogens on the health of both captive and wild populations.

In addition to the health implications of viral infections, viruses may also impact host evolution (*Forterre & Prangishvili, 2009*; *Villarreal & Witzany, 2010*). Viruses often exchange genetic material with their hosts, and they are implicated in several major evolutionary transitions, for example the "invention" of DNA polymerase (*Filée et al., 2002*; *Forterre, 2006*). It is unclear what role viruses may be playing in the evolution of extant lineages, but by exploring how host and viral lineages evolve, we may begin to address this question.

Adenoviruses (AdVs) are a group of non-enveloped, double-stranded DNA viruses that occur as pathogens throughout vertebrates, including humans. Although most known strains occur in mammals, recent work has shown that they are widespread among squamate reptiles as well (*Marschang, 2011*; *Ariel, 2011*; *Szirovicza et al., 2016*). AdVs have been identified in more than 30 squamate species to date, and have been shown to cause morbidity and mortality in numerous captive-bred pet trade species (*Frye et al., 1994*; *Kim et al., 2002*; *Marschang et al., 2003*; *Pasmans et al., 2008*; *Moormann et al., 2009*; *Ball et al., 2014*). However, not all reptile species that harbor adenovirus show adverse health effects (*Jacobson & Kollias, 1986*; *Ogawa, Ahne & Essbauer, 1992*; *Parkin et al., 2009*; *Doszpoly et al., 2013*; *Kubiak, 2013*); the pathogenicity of adenovirus infection across reptiles is unclear.

Based on work in non-reptilian taxa, AdVs are generally thought to exhibit host-specificity, and to co-evolve with host taxa (*Wellehan et al., 2004*; *Harrach & Benko, 2007*; *Papp et al., 2009*). However, host switching has also occurred. Phylogenetic study of AdVs present in deeply divergent tetrapod hosts suggest that AdVs in the genus *Atadenovirus* have been transferred between reptiles and birds, marsupial mammals, and ruminant mammals (*Farkas et al., 2002*; *Farkas, Harrach & Benkő, 2008*). In addition, within reptiles, distantly related taxa have been shown to harbor identical AdVs, suggesting more recent transfer of AdVs between divergent host lineages (*Hyndman & Shilton, 2011*; *Marschang et al., 2003*).

By contrast, across shallower host divergences AdV infection appears to be relatively species-specific. AdV sequences isolated from conspecific individuals tend to form monophyletic clades, such that members of the same host species are generally infected by the same (or very similar) AdV strain (*Papp et al., 2009*; *Doszpoly et al., 2013*). This pattern holds for both wild and captive squamates (*Szirovicza et al., 2016*), but it is important to note that the vast majority of this evidence comes from captive pet-trade animals, where pathogen exposure, transmission, and disease dynamics are likely to differ from wild populations.

To understand the evolutionary dynamics between reptile AdVs and their hosts, as well as the general pathology of adenovirus infection, it is important to sample wild individuals. To date, only two studies have tested for AdV in wild-caught squamate specimens. The first examined 56 individuals in three species of limbless

squamates, and found no adenovirus in any of their samples (*Schmidt et al., 2014*). The second examined 318 individuals from eight lizard species, and found AdV in 22 samples representing three host species (two Lacertids and one Amphisbaenid; 6.9% prevalence) (*Szirovicza et al., 2016*). These AdV strains all fell within the *Atadenovirus* genus. The individuals that tested positive did not show any signs of morbidity. This low overall prevalence and lack of clinical or disease signs suggests that AdV may not be a significant pathogen in wild populations of these species. However, the mode of transmission, prevalence, and source of AdVs in the environment are still unknown, and the dynamics and effects of AdV infection may vary in different squamate clades.

In the present study, we explore the phylogenetic history and prevalence of adenovirus in six disparate native populations of the brown anole (*Anolis sagrei*) from the Bahamas and the Cayman Islands. *A. sagrei* is a widespread species that occurs on many islands throughout the Caribbean, and provides an excellent model system in which to study AdV prevalence and evolution within a single host species. We opportunistically sampled AdV in deceased captive *A. sagrei* individuals that were collected for a breeding experiment, as well as fecal samples collected in situ from one source population. We sequenced a portion of the adenovirus polymerase gene and inferred a phylogenetic tree including known AdVs to address the following clinical and evolutionary hypotheses: (a) lab colony morbidity and mortality was associated with AdV infection, (b) individuals acquired AdV from the wild, not from other individuals in the lab colony, (c) AdV has co-diversified in parallel with its *A. sagrei* hosts, and (d) AdV from *A. sagrei* forms a clade with AdV isolated from other *Anolis* species.

## MATERIALS AND METHODS

In 2016 we imported a total of 900 *A. sagrei* lizards from six islands in the Bahamas and the Cayman Islands (Abaco, Bimini, Conception Island, Staniel Cay, Little Cayman and Cayman Brac), composed of 75 adult individuals of each sex per island (Fig. 1). Permission for collection and export of animals was granted by the BEST Commission and the Ministry of Agriculture in the Bahamas and the Cayman Islands Department of the Environment. Individuals were captured in natural forested habitat either by hand or using a standard noosing technique, and were transported to an animal care facility at Harvard University in Cambridge, MA. Anoles were housed in custom acrylic cages containing several wooden dowel perches, artificial foliage, and sterilized soil substrate. Cages were misted twice daily to provide drinking water. Animals were fed crickets dusted with calcium and multivitamin powders ad libitum three times weekly. Room temperature was maintained at 84 F. Photoperiod was cycled to mimic natural conditions (lights were on from 7 am to 9 pm). Males were housed individually, while females were housed in groups of 2–3. In female cages a 500 ml polypropylene cup filled with moistened vermiculite was placed as egg-laying substrate. Animals from different populations were housed separately. All cages and cage dressings were sterilized prior to use and steps were taken to minimize the possibility of cross-contamination between populations (e.g., gloves were changed between handling animals from different populations). All animal

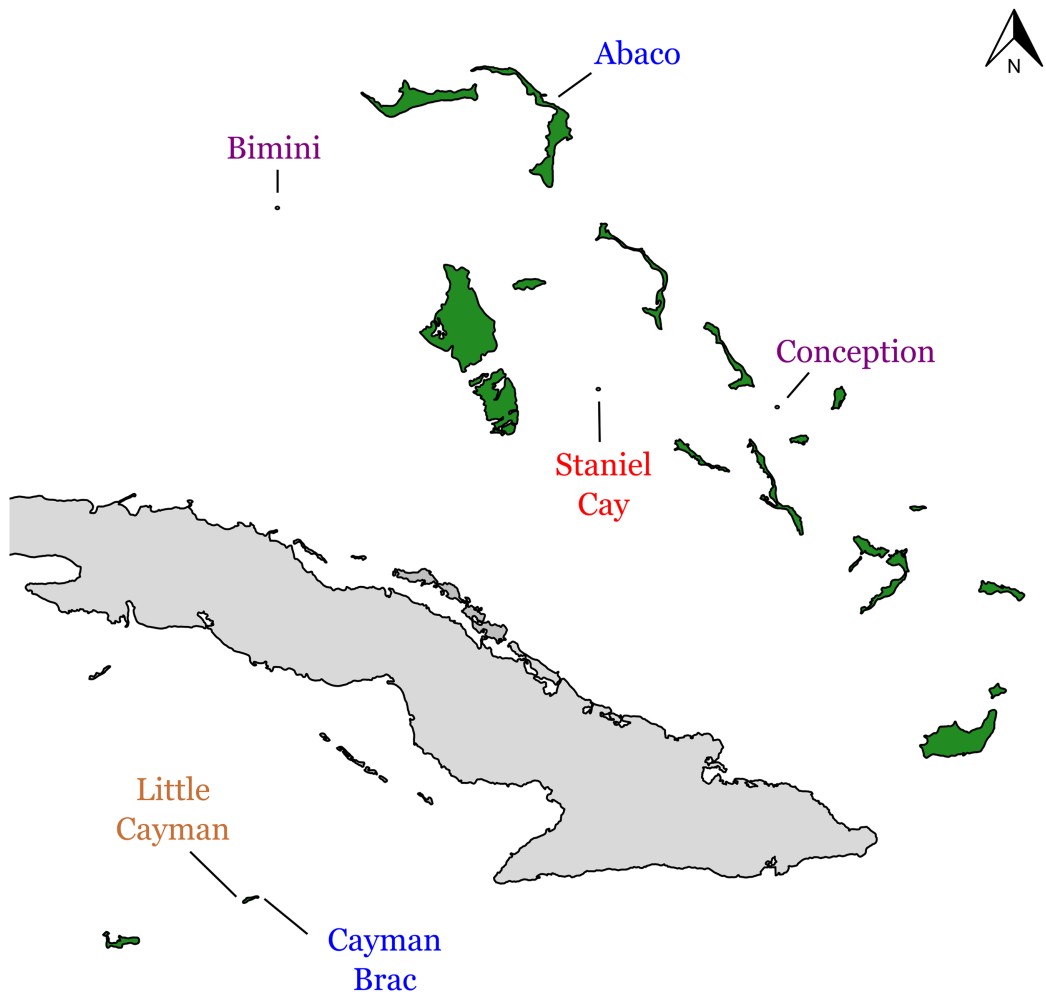

**Figure 1 Geographic sources of adenovirus-infected individuals.** *Anolis sagrei* was collected from four islands in the Bahamas and two of the Cayman Islands. Color of island names corresponds to adenovirus clades (Fig. 2). Figure was generated using the package mapdata (*Becker, Wilks & Brownrigg, 2016*) in R (*R Core Team, 2017*).             

care procedures were approved by the Harvard University Institutional Animal Care and Use Committee (Protocol 26-11).

In late May and early June 2016, we noticed an increase in morbidity and mortality of captive animals, particularly among females. Clinical signs of disease included visible weight loss and lethargy. Animals exhibiting disease signs were isolated and monitored but none recovered once these signs were observed. The transition from a wild environment to the laboratory is stressful; based on previous *Anolis* colony-based studies, mortality associated with this transition is expected to be about 10%. In the present study, over the course of seven months after colony establishment, 14% of captive animals died, which was greater than the expected baseline. Previous experience with clinical signs of adenoviral infection (*Ascher et al., 2013*) caused us to suspect an adenovirus infection as a potential cause of colony mortality. We therefore decided to test for adenovirus infection in a subset of the deceased individuals using PCR. We then sequenced the AdV PCR

**Table 1 Population sampling.**

| Island | No. individuals sampled | No. adenovirus positives |
|---|---|---|
| Abaco, Bahamas | 6 | 2 |
| Bimini, Bahamas | 7 | 1 |
| Conception, Bahamas | 4 | 2 |
| Staniel Cay, Bahamas | 9 | 2 |
| Cayman Brac, Cayman Islands | 7 | 1 |
| Little Cayman, Cayman Islands | 5 | 2 |
| Total | 38 | 10 |

Note:
Numbers of deceased individuals sampled and numbers of adenovirus positives from each study population. Total adenovirus prevalence is 26%.

product for all AdV-positive individuals, and used those sequences to construct a phylogenetic tree of AdVs in *A. sagrei*.

We extracted DNA from 38 deceased lizards across the six populations, as well as from 10 fecal samples that were opportunistically collected in the field from Staniel Cay, to test whether any adenovirus present in the lab could be shown to originate from a wild source (Table 1; Fig. 1). We obtained tissue samples from the deceased lizards by removing approximately 20 mg of abdominal tissue with sterile disposable razor blades. We extracted DNA from these samples using the E.Z.N.A. Tissue DNA Kit (Omega Bio-Tek, Norcross, GA, USA). We collected the fecal samples from live animals in the field in Staniel Cay only. In this population, lizards were gently palpated upon capture to induce defecation, and the fecal bolus was wiped directly from the cloaca into a vial containing one ml RNALater (ThermoFisher Scientific, Waltham, MA, USA). DNA was extracted from fecal samples using the PureLink Microbiome DNA Purification Kit (ThermoFisher Scientific, Waltham, MA, USA).

We used previously published degenerate PCR primers (Wellehan et al., 2004) to perform a nested PCR to amplify an approximately 320 bp fragment of the adenoviral DNA polymerase gene from our 48 samples and one positive control from a previous study (Ascher et al., 2013). We performed 25 μl reactions with 12.875 μl diH$_2$O, 2.5 μl of primers polFouter and polRouter at two μM, 2.5 μl of EconoTaq Buffer with Mg (Lucigen, Middleton, WI, USA), 2.5 μl of 0.5 mM dNTP solution, 0.125 μl of EconoTaq DNA Polymerase (Lucigen, Middleton, WI, USA), and two μl of template DNA. We amplified the 320 bp coding region of highly conserved adenovirus polymerase gene using the following thermocycler protocol: 94 °C for 720 s, followed by 45 cycles of 94 °C for 30 s, 46 °C for 60 s, and 72 °C for 60 s, followed by 72 °C for 10 min. We purified the PCR products using the E.Z.N.A. Cycle Pure Kit (Omega Bio-Tek, Norcross, GA, USA). We performed the second PCR using the same protocol, with two μl of the purified PCR product as template and using polFinner and polRinner primers. We then visualized the DNA using a 1.5% agarose gel stained with GelRed nucleic acid gel stain (Biotium, Inc., Fremont, CA, USA). Successful PCR products were sent to the CCIB DNA Core for purification and Sanger sequencing in both directions using the polFinner and polRinner primers (Cambridge, MA, USA).
**Table 2 Adenovirus sequences included in alignments.**

| Clade | Adenovirus isolate | GenBank accession # | Reference |
|---|---|---|---|
| Polychrotidae | ADV20 *Anolis sagrei* Staniel | MH558566 | New |
| Polychrotidae | ADV22 *Anolis sagrei* Staniel | MH558567 | New |
| Polychrotidae | ADV27 *Anolis sagrei* Little Cayman | MH558568 | New |
| Polychrotidae | ADV29 *Anolis sagrei* Little Cayman | MH558569 | New |
| Polychrotidae | ADV10 *Anolis sagrei* Bimini | MH558563 | New |
| Polychrotidae | ADV15 *Anolis sagrei* Conception | MH558565 | New |
| Polychrotidae | ADV13 *Anolis sagrei* Cayman Brac | MH558564 | New |
| Polychrotidae | ADV2 *Anolis sagrei* Abaco | MH558562 | New |
| Gekkonidae | Gekkonid AdV-1 | AY576681 | *Wellehan et al. (2004)* |
| Scincidae | Scincid AdV-1 | AAS89698 | *Wellehan et al. (2004)* |
| Amphisbaenidae | Amphisbaenian AdV-1 | KT950887 | *Szirovicza et al. (2016)* |
| Lacertidae | Lacertid AdV-1 | KT950888 | *Szirovicza et al. (2016)* |
| Lacertidae | Lacertid AdV-2.20131 | KT950885 | *Szirovicza et al. (2016)* |
| Lacertidae | Lacertid AdV-2.20132 | KT950886 | *Szirovicza et al. (2016)* |
| Agamidae | Agamid AdV-1.C1 | ACI28428 | *Parkin et al. (2009)* |
| Agamidae | Agamid AdV-1.A1 | AAS89694 | *Wellehan et al. (2004)* |
| Iguanidae | Chameleon AdV-1 | AY576679 | *Wellehan et al. (2004)* |
| Polychrotidae | *Anolis* AdV-2 (*distichus ravitergum*) | KC544016 | *Ascher et al. (2013)* |
| Polychrotidae | *Anolis* AdV-1 (*distichus ignigularis*) | KC544015 | *Ascher et al. (2013)* |
| Polychrotidae | *Anolis carolinensis* AdV-3 | KF886534 | *Ball et al. (2014)* |
| Agamidae | Green striped tree dragon AdV-1 | KF886532 | *Ball et al. (2014)* |
| Serpentes | Snake AdV-2 | FJ012163 | *Garner et al. (2008)* |
| Eublepharidae | Eublepharid AdV-1 | AY576677 | *Wellehan et al. (2004)* |
| Agamidae | Western bearded dragon AdV-1 | HQ005514 | *Hyndman & Shilton (2011)* |
| Helodermatidae | Helodermatid AdV-1 | AY576680 | *Wellehan et al. (2004)* |
| Serpentes | Snake AdV-3 | ACH91015 | *Garner et al. (2008)* |
| Serpentes | Snake AdV-GER09 | ADT91320 | *Abbas et al. (2011)* |
| Serpentes | Snake AdV-GER09 | ADT91320 | *Abbas et al. (2011)* |
| Mammalia | Human AdV-12 | AY780216 | *Chmielewicz et al. (2005)* |
| Mammalia | Human AdV-12 | M14785 | *Shu et al. (1986)* |
| Mammalia | Human AdV-3.NHRD.1276 | AY599836 | *Lin et al. (2006)* |
| Mammalia | Human AdV-B Guangzhou01 | DQ099432 | *Zhang et al. (2006)* |
| Mammalia | Human AdV-16.CH79 | AY601636 | *Lin et al. (2006)* |
| Mammalia | Human AdV-7.NHRC.1315 | AY601634 | *Lin et al. (2006)* |
| Mammalia | Simian AdV-33 | FJ025908 | *Roy et al. (2009)* |
| Mammalia | Human AdV-11.SGN1222 | FJ597732 | Unpublished |
| Mammalia | Pygmy marmoset AdV | HM245776 | *Gál et al. (2013)* |
| Mammalia | Bat AdV-isolate 1391 | GU226964 | *Li et al. (2010)* |
| Mammalia | Porcine AdV-5 | AF289262 | *Nagy, Nagy & Tuboly (2001)* |
| Mammalia | Bovine AdV-2 | AF252854 | *Salmon, Esford & Haj-Ahmad (1993)* |
| Mammalia | Tree shrew AdV-1 | AF258784 | *Darai et al. (1980)* |
| Mammalia | Bovine AdV-3 | AF030154 | *Lee et al. (1998)* |

| Clade | Adenovirus isolate | GenBank accession # | Reference |
|---|---|---|---|
| Mammalia | Bat AdV-isolate 1213 | GU226951 | *Li et al. (2010)* |
| Mammalia | Bat AdV-isolate 1497 | GU226967 | *Li et al. (2010)* |
| Mammalia | Canine AdV-2 | U77082 | *Shibata et al. (1989)* |
| Mammalia | Canine AdV-1 | Y07760 | *Morrison, Onions & Nicolson (1997)* |
| Mammalia | Bat AdV-2.PPV1 | JN252129 | *Sonntag et al. (2009)* |
| Mammalia | Bat AdV-1069 | GU226963 | *Li et al. (2010)* |
| Mammalia | Bat AdV-1282 | GU226960 | *Li et al. (2010)* |
| Mammalia | Bovine AdV-D | NC002685 | *Dán et al. (2001)* |
| Mammalia | Ovine AdV-7 | U40839 | *Vrati et al. (1995)* |
| Mammalia | Murine AdV-3 | NC012584 | *Klempa et al. (2009)* |
| Amphibia | Frog AdV-1 | NC002501 | *Davison, Wright & Harrach (2000)* |
| Aves | Meyers parrot AdV-1 | AY644731 | *Wellehan et al. (2005)* |
| Aves | Fowl AdV-5.1422 | DQ159938 | *Hanson et al. (2006)* |
| Aves | Plum headed parakeet AdV-1 | EU056825 | *Wellehan et al. (2009)* |
| Aves | Great tit AdV-1 | FJ849795 | *Kovács et al. (2010)* |
| Aves | Avian AdV-EDS | Y09598 | *Hess, Blöcker & Brandt (1997)* |
| Testudines | Sulawesi tortoise AdV-1 | EU056826 | *Rivera et al. (2009)* |
| Testudines | Box turtle AdV-1 | EU828750 | *Farkas & Gál (2009)* |
| Testudines | Box turtle AdV-R08-207 | JN632579 | *Doszpoly et al. (2013)* |
| Testudines | Pancake tortoise AdV-R08-227 | JN632575 | *Doszpoly et al. (2013)* |
| Testudines | Red-eared slider turtle AdV-V16 | JQ801339 | *Doszpoly et al. (2013)* |
| Actinopterygii | White sturgeon AdV | Not accessioned in GenBank | *Benkő et al. (2002)* |

We aligned the sequences we collected with 128 known adenovirus sequences drawn from public databases. We assembled and aligned sequences by eye using Geneious v9.1.5 (*Kearse et al., 2012*). After alignment, we translated the nucleotide sequences to amino acids. We did not use the raw nucleotide alignment for phylogenetic analyses because preliminary analyses showed it to be saturated (Fig. S1); therefore all further analyses are based on the amino acid alignment (Table 2).

We used PartitionFinder v1.1.1 (*Lanfear et al., 2012*) to find the best fitting model of molecular evolution for the amino acid alignment. We inferred a phylogenetic tree using the Bayesian Metropolis-coupled Markov Chain Monte Carlo algorithm in MrBayes v3.2.6 (*Ronquist et al., 2012*) using an LG+I+G model. We ran MrBayes for 300 million generations with one cold and three heated chains, sampling every 10,000 generations. White sturgeon AdV (*Benkő et al., 2002*) and Box Turtle AdV (*Farkas & Gál, 2009*) were selected as outgroups. After evaluation, the first 50% of trees were removed as burn-in. Analyses were run on the CIPRES server (*Miller, Pfeiffer & Schwartz, 2010*). We assessed convergence of MCMC runs using RWTY v1.0.1 (*Warren et al., 2017*), with particular attention to (1) stationarity in the posterior distribution of parameter estimates within runs and (2) correlation in the magnitude of node support (posterior probability) among independent runs.

## RESULTS

Using nested PCR, we tested 38 deceased lizards from six island populations for adenovirus infection, and found 10 positives (prevalence = 26%; Table 1). We also tested 10 field-collected fecal samples from one island (Staniel Cay), and found one positive sample (prevalence = 10%). These observations represent the first case of AdV identified in a wild *Anolis* lizard, the third case of AdV infecting *Anolis* generally (*Ball et al., 2014*; *Ascher et al., 2013*), and the third case of AdV reported from a wild squamate species (*Szirovicza et al., 2016*). We successfully sequenced and assembled adenovirus sequences from each the 11 positive samples and inferred the phylogeny of the resulting amino acid sequences and previously published Adenovirus polymerase sequences.

Our resulting phylogeny overall is consistent with recently constructed adenovirus trees (Fig. 2). It recovers monophyletic viral genera *Aviadenovirus*, *Mastadenovirus*, *Atadenovirus*, *Siadenovirus* (*Harrach et al., 2011*), and the recently proposed "Testudine-infecting AdV" genus (*Doszpoly et al., 2013*). Within the primarily reptile-infecting *Atadenovirus* clade, our tree shows support for previously established AdV subclades (e.g., Snakes + Eublepharids + Helodermatids, Anolis + chameleon, Agamids), and similar overall structure (i.e., Gekkonid AdV as sister to the remaining *Atadenovirus*) (*Wellehan et al., 2004*; *Ascher et al., 2013*; *Ball et al., 2014*; *Szirovicza et al., 2016*). Posterior probabilities are highest for shallower nodes, while deeper nodes are more poorly resolved; the monophyly of the AdV genera is well-supported, but the relationships between the genera are equivocal in our analysis.

Five of the 11 *A. sagrei* AdV sequences in the present study form a monophyletic group with previously sequenced *Anolis* AdVs, as predicted. The remaining six sequences form two distinct clades. The first contains three sequences from two islands, Cayman Brac and Abaco, and is sister to the green striped tree dragon AdV within the genus *Atadenovirus*. The other clade contains three sequences from Staniel Cay, one of which was obtained from a field-collected fecal sample. This clade is quite distant from the rest of the *Anolis* AdV sequences; it is entirely outside of the reptile-infecting *Atadenovirus* genus, and is instead sister to the mammal associated *Mastadenovirus* clade.

## DISCUSSION

The aim of our study was to address four hypotheses related to the pathogenicity and evolution of adenovirus in *A. sagrei*: (a) observed lab morbidity and mortality was associated with AdV infection, (b) individuals acquired AdV from the wild, (c) AdV has co-diversified in parallel with its *A. sagrei* hosts, and (d) AdV from *A. sagrei* forms a clade with AdV derived from other *Anolis* species.

We expected that if AdV infection were the main cause of morbidity and mortality, the majority of our samples would test positive for AdV in abdominal tissue. We found that AdV was present in 26% of samples, suggesting that AdV was not the primary cause of mortality in our population (Table 1). In addition, the AdV-positive field fecal sample came from an individual who was quite healthy in the lab, and which survived the mortality event, lending further support to this conclusion. However, although the prevalence was relatively low, the fact that we identified AdV in several individuals

**Figure 2 Phylogenetic relationships among adenoviruses inferred using amino acid sequences.** Novel sequences are shown in gray boxes. Text color of clades corresponds to the island from which lizards were collected (Fig. 1). (*) denotes a sequence that was present in a fecal sample acquired in the field. Parenthetical numbers next to locality names represent adenoviruses that were represented by multiple identical viral sequences from different host individuals. Node support is presented as posterior probabilities (PP). Viral genera are noted on the right. Scale bar shows number of amino acid substitutions per site. "Testudine-infecting AdVs" comprise a putative new genus of adenovirus (*Doszpoly et al., 2013*).

suggests that while AdV likely did not cause the death of all the deceased colony animals, it may have been a cofactor.

These results should be interpreted with some caution. Due to the nature of the samples (abdominal tissue of deceased animals), both false positives and false negatives are possible. Deceased individuals had undergone various degrees of minor decomposition prior to sample extraction, so while we were able to sample an approximately equal mass of tissue from each individual, we cannot be sure that the samples contained the same internal organs in the same proportions. Therefore, it is possible that some lizards which tested negative in fact harbored adenovirus in organs that we did not sample. We extracted a rather large portion of abdominal tissue to mitigate this possibility, but nonetheless we cannot rule out that sampling had an effect on adenovirus detection. By contrast, it is also possible that individuals which tested positive did not harbor active infections, and instead were passive carriers which had ingested inactive adenovirus from an environmental source. However, because most individuals that tested positive had genetically distinct AdVs, we believe most if not all of these animals harbored independent infections.

Interestingly, adenovirus was found in more females than males (8/19 and 2/9, respectively), but the difference was not significant ($\chi^2 = 0.10$, $p = 0.75$). Of the eight females that tested positive, only two were housed in the same cage, suggesting that the higher number of female infections cannot be explained by cohabitation alone (all males were housed singly). However, adenovirus infection in *A. sagrei* is not always lethal. One AdV-positive Staniel Cay field sample came from an individual that showed no signs of disease in the field nor after transfer to the vivarium. It is therefore possible that some surviving females were infected with AdV but showed no signs of disease and instead acted as carriers, transmitting AdV to their cage-mates and leading to higher incidence of AdV infection among females overall. Due to our lizard care design, we are unable to determine whether the slight female bias in mortality is due to increased transmission of AdV in communal cages, or whether females are inherently more susceptible.

We found strong evidence that the infections present in the lab were a result of independent wild infections, not horizontal transfer between captive animals from different populations. We successfully sequenced AdV from one field-acquired fecal sample, and this sequence was identical to a sequence obtained from a separate captive animal that came from the same source population (Staniel Cay) (Fig. 2). These two sequences were together sister to another AdV sequence present in a different individual from the same population. This is the first record of a field strain subsequently and independently being identified in the lab, and provides strong evidence that the population harbored AdV in the wild, and transferred it into the lab environment. Further, in the populations in which we found two or more individuals infected (Staniel Cay, Little Cayman, Conception, and Abaco), the AdV sequences from these individuals were either identical (Conception, Abaco) or sister (Staniel Cay, Little Cayman). In addition, while we cannot be certain that no transmission occurred in lab (e.g., between Bimini and Conception animals, or between Cayman Brac and Abaco, which shared identical AdV

sequences), the fact that there are four distinct clades present in our lab colony suggests that *A. sagrei* is infected by diverse AdV strains in the wild.

The 11 AdV sequences we identified from the six different island populations of *A. sagrei* (Fig. 1) form four distinct clades (Fig. 2). These clades fall within three separate adenovirus lineages, and represent a surprisingly diverse set of AdVs for one host species. Two of these clades (Little Cayman and (Bimini + Conception)) fall within a clade of previously identified *Anolis* AdVs, supporting our hypothesis that *Anolis* lizards harbor similar strains of AdV. However, by contrast, the other two clades (Staniel Cay and Cayman Brac + Abaco) were more divergent than expected in host-virus co-diversification scenario. One of the remaining clades (Cayman Brac + Abaco) was nested within a separate clade of non-anoline squamate AdVs, and was more closely related to AdV strains found in species of snakes (Serpentes), Helodermatids, Eublepharids, and Agamids (green striped tree dragon and Western bearded dragon). The last *A. sagrei* AdV clade (Staniel Cay), was the most divergent, and fell entirely outside of reptilian AdVs, instead appearing as sister to the genus *Mastadenovirus*, which primarily infect mammals.

*Anolis sagrei* are unique among reptiles studied thus far in having numerous diverse AdV strains infecting one species. For example, a study of multiple individuals of *Heloderma* and *Pogona* showed little to no diversity in AdV strains present (*Papp et al., 2009*). In a study of several turtle species, multiple individuals of one species (*Trachemys scripta*) had numerous distinct adenovirus strains, but they were all closely related, forming a clade within the testudine adenovirus group (*Doszpoly et al., 2013*). However, as mentioned above, these studies focused on captive-bred pet trade animals. Captive-bred animals are separated from potential wild sources of pathogens for generations, so any pathogen that enters captive-bred lines is likely to be the result of horizontal transmission from other captive animals, rather than natural acquisition. It is therefore likely that the current picture of AdV diversity in squamates is limited by the nature and condition of study subjects, and in the wild, AdV is much more diverse than we realize.

In summary, our results show evidence for both host-virus co-diversification and host switching within a single host species. Highly divergent viral clades infected closely related host populations, such that the AdV present in one population (Staniel Cay) is sister to mammalian AdVs, while others, for example (Bimini + Conception) are nested within the reptilian *Atadenovirus* clade, suggesting host switching between divergent host lineages. However, by contrast, sequences from three of the six *A. sagrei* populations (Bimini, Conception, and Little Cayman) form a clade with previously identified anoline AdVs, suggesting a degree of host-specificity and potential co-diversification between *Anolis* species and the AdVs that infect them.

There are several possible explanations for this pattern. First, we sampled populations occurring in distant localities. These populations may occur in distinct species communities, and individuals on one island may be exposed to different viruses through other community members. This explanation seems somewhat unlikely because the most divergent viral strain (from Staniel Cay) was found on an island that is ecologically very similar to the nearby Bahamian islands of Bimini and Conception, which instead harbored AdVs that were phylogenetically nested within the squamate AdV clade.

These different populations share the same reptile community members, and are likely to co-occur with the same amphibian, avian and mammalian species as well. However, we did not quantify the abundance or diversity of other species, so it is possible that interactions between different community members varies by island.

Alternatively, it is possible that both AdV diversity and AdV host switching are simply much higher than expected in the wild. The relative lack of diversity seen in captive species may be primarily an effect of their captivity; without exposure to other conspecific and heterospecific species in the wild, these animals have no chance to encounter other strains of AdV, and are therefore unlikely to reflect the natural diversity of AdV found in natural populations or communities. In addition, it is possible that AdV is more efficient at switching between hosts than previously appreciated, such that what was previously viewed as surprising host-switching events between deeply divergent hosts may in fact be quite common. Given that AdV has only been sampled in a small number of wild species and populations to date, we are currently unable to determine which of these explanations is more likely.

In the current study, we sought to explore the disease dynamics and evolutionary history of AdV within a single widespread reptilian species, *A. sagrei*, by opportunistically sampling deceased individuals from a recently collected captive colony. We found limited evidence that AdV was the primary cause of mortality in our colony, although it may have been a contributing factor. We found a surprising degree of AdV diversity among the six host populations we sampled, which provides evidence for both host switching and virus-host co-diversification across different host populations. Further studies of presence and diversity of AdV strains in wild reptiles will be important for understanding the short- and long-term evolutionary and ecological dynamics of this host-virus relationship. It will also be important to screen and monitor both captive and wild reptile populations to understand the prevalence and pathogenicity of adenovirus infection, and its overall impact on wildlife health and disease. Future work isolating various adenovirus strains and performing infection trials would also be extremely informative, and would greatly contribute to our understanding of the pathology and mode of action of adenovirus infection in different reptile species.

## CONCLUSION

Adenoviruses infect a wide range of hosts, and have been found in many captive reptile species, as well as a small number of wild species. They have undergone recent and ancient host transfers multiple times, but the pathogenicity and evolutionary history of reptile AdVs remains poorly understood. We addressed several hypotheses relating to adenoviral pathogenicity and evolution within a single widespread reptile species, *A. sagrei*, using an opportunistic sampling of deceased captive individuals and field-collected fecal samples. We found that adenovirus was likely not the primary cause of observed morbidity and mortality in our colony, but was present in a subset of our samples, and may have been a cofactor. Future work explicitly testing the pathogenicity of these viruses in non-captive-bred species would be highly informative. We also identified four separate adenovirus clades present in our six populations of *A. sagrei*, several of which

are distantly related, suggesting that AdVs may undergo frequent host switching across both deep and shallow host divergences, and contrasting with a host-virus co-speciation model. Further work investigating adenovirus prevalence and evolution in wild reptile populations will be important for understanding broader patterns of host-adenovirus evolution.

## ACKNOWLEDGEMENTS

We would like to thank the BEST Commission and the Ministry of Agriculture in the Bahamas and the Cayman Islands Department of the Environment for granting permission to sample and export *A. sagrei*. We would like to thank Shea Lambert, Alexis Harrison, Inbar Maayan, Shannan Yates, Nicholas Herrmann, Kristin Winchell, Dan Scantlebury, R. Graham Reynolds for their help in the field, as well as Cory Hahn, Matt Gage, Jeff Breeze, and Analisa Shields-Estrada for their assistance with the lab colony. We would also like to thank Jonathan Losos for logistical, financial, and intellectual support. We thank Fernando Spilki and two anonymous reviewers for their comments on a previous version of this manuscript.

### Funding

This work was supported by the National Science Foundation (NSF DEB #1500761 to Anthony J Geneva and DGE #1144152 to Sofia Prado-Irwin) and the John Templeton Foundation (to Jonathan Losos). The funders had no role in study design, data collection and analysis, decision to publish, or preparation of the manuscript.

### Grant Disclosures

The following grant information was disclosed by the authors:
National Science Foundation (NSF DEB #1500761 to Anthony J Geneva and DGE #1144152 to Sofia Prado-Irwin).
John Templeton Foundation (to Jonathan Losos).

### Competing Interests

The authors declare that they have no competing interests.

### Author Contributions

- Sofia R. Prado-Irwin performed the experiments, analyzed the data, contributed reagents/materials/analysis tools, prepared figures and/or tables, authored or reviewed drafts of the paper, approved the final draft.
- Martijn van de Schoot performed the experiments, analyzed the data, approved the final draft.
- Anthony J. Geneva conceived and designed the experiments, analyzed the data, contributed reagents/materials/analysis tools, authored or reviewed drafts of the paper, approved the final draft.

## Animal Ethics

The following information was supplied relating to ethical approvals (i.e., approving body and any reference numbers):

All animal care procedures were approved by the Harvard University, Faculty of Arts and Sciences Institutional Animal Care and Use Committee (IACUC) (Protocol 26-11).

## Field Study Permissions

The following information was supplied relating to field study approvals (i.e., approving body and any reference numbers):

Permission for collection and export of animals was granted by the BEST Commission and the Ministry of Agriculture in the Bahamas and the Cayman Islands Department of the Environment.

## Data Availability

The adenovirus polymerase gene sequences described here are available as Supplemental Files as well as at GenBank: MH558562–MH558569.

## Supplemental Information

Supplemental information for this article can be found online at http://dx.doi.org/10.7717/peerj.5521#supplemental-information.

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
