# Peer review of "Detection and phylogenetic analysis of adenoviruses occurring in a single anole species"

_PeerJ, doi:10.7717/peerj.5521_

## Round 0.1 · original submission · Minor Revisions

Please give special attention to the corrections needed on the phylogenetic analysis and important references missing.

Reviewer 1 ·

Basic reporting

This is an interesting manuscript about the detection of novel adenoviruses by PCR in wild-living anole lizards. The aim was to find out if individuals of the same species (Anolis sagrei), originating from different, well-separated geographical regions, may harbour identical, closely related or rather divergent adenoviruses. Secondly, the authors wanted to examine if the somewhat increased mortality that had been observed in the newly established captive lizard colony, could be the consequence of adenovirus infection or not. The paper in general is well written, the majority of the methods used to clarify the questions are adequate and this is true for the conclusions as well. A minor to medium-level revision would help increasing the scientific significance of the paper. Specific points are listed below.
First of all I would suggest changing the title to a more modest one, such as e.g. “Detection and phylogenetic comparison of adenoviruses occurring in a single anole species”, because no real evidences or conclusions regarding the evolution of adenoviruses are thoroughly discussed.

Experimental design

The authors began to screen the dead lizards a bit late, so that only a portion (less than one third) of them could be examined. Nonetheless, I think the overall positivity rate of 26% can be considered high. Adenoviruses tend to establish persistent infection, i.e. after the resolution of the disease, caused by the primary infection, the virus remains in the host without any clinical signs for long periods. In case of decreased immunocompetence of the host, the virus may start to replicate, sometimes resulting in disease again, or just being shed to the environment.
In the described cases, it is hard to comprehend the pathologic role of the detected viruses, since the exact nature of the samples is not specified. The composition the 20 mg of “abdominal tissue” (line 117) might result in huge differences depending on the inclusion (or exclusion) of certain parenchymal organs such as liver, spleen, and kidney. I know that rapid autolysis can prevent the exact determination of the sample, but still it might have a major influence on the outcome of the PCR tests and therefore it should not be left out of consideration when counting the possibility of false negative tests. On the other hand, the presence of virus in the intestinal content (or cloacal swabs) does not necessarily signals active infection, as it may as well originate from a passive carrier status of the host e.g. the virus was ingested and can be detected by nested PCR even in lack of active replication in the given animal. I would recommend a more cautious wording after consideration of the possibilities mentioned above.

Validity of the findings

Unfortunately, the newly-gained sequences are not released, and even the GenBank accession numbers seem not to be assigned yet although it is a prerequisite for acceptance at most journals. I would guess it is so at PeerJ as well. Moreover, the scale of the phylogenetic tree on Figure 2 is missing thus making the comprehension of differences difficult. The majority of the newly detected sequences fit into the genus Atadenovirus and appear close to AdV sequences derived from closely related hosts. The only exceptions are the three sequences originating from the lizards at Staniel Island. Their closest hits in the blast application (if it was run at all) would have been of interest. Mastadenoviruses occurring in more ancient mammals (rodents and especially mice) should have been included into the calculation. This could have helped clarifying the question if possible genus affiliation of the AdV sequences derived from lizards of the Staniel Cay Island.
An explanation for the higher mortality and AdV prevalence, among female lizards compared to males, could be the common caging (instead of individual one) that provides better conditions for the horizontal virus transmission. It would have been interesting to know, how many of the affected animal were kept in shared cages.
The authors seem to be very familiar with the literature of reptilian adenoviruses. Nonetheless, one relevant paper (see below) is badly missing from the list of references. Although this paper is about AdVs found in non-squamate reptiles (namely in different sliders and turtles), yet it presents several genomic variants of the same virus, sometimes present concurrently in the same sample. In this work the sequences of the same PCR fragment from the viral DNA polymerase gene were examined, so that the results can easily be compared to those of the present manuscript.

Additional comments

Typing errors and word usage mistakes:
Line 1: adenovirus is an adjective here neither capital letters, nor italics are needed. The same applies to “adenovirus” in lines 154, 159, 161, 179 & 208.
Lines 72, 104 (twice), 105, 109: I would recommend to use “clinical or disease signs”, as “symptom” refers to a subjective feeling of the patient. This word is not really adequate when speaking about animals.
Lines 117 & 121: The numbers should be separated from the measurement units by spaces.
Line 133: To avoid redundancy, the word “reaction” should be deleted.
In the first lien of legends for Figure 2 (after line 169): “adenovirusus” should be corrected. Also, a scale should be provided.
Figure 2: At least one (out of the three) murine adenovirus should also be included. The vertical lines marking the genera have been mixed up. From the genus Aviadenovirus, only two representatives are shown, whereas four viruses from the genus Siadenovirus. Therefore this latter one should be marked with the longer line. The spelling (regarding capitals and italics) of the genus names is correct. In our experience, the Bayesian analysis is not the optimal choice in case of such short sequences.
Lines 174, 181 & 196: I would recommend changing “isolated” to “obtained” or “derived”. Isolate implies that the virus is available as a strain suitable for in vitro replication in cell culture. (I have replaced the word “strain” with “virus” in the title for the same reason.)
Lines 330 & 331: Please double-check the authors’ list of this reference.
I recommend addition of the following reference:
Doszpoly A et al. (2013) Partial characterization of a new adenovirus lineage discovered in testudinoid turtles. Inf. Genet. Evol. 17:106‒112.

Reviewer 2 ·

Basic reporting

No comment.

Experimental design

No comment.

Validity of the findings

No comment.

Additional comments

The manuscript has clear language and is well written. Minimal corrections are suggested. Only as a suggestion for future work, if the research group is interested, inoculate the samples (both positive and negative) in cell line for analysis of infectivity and viral isolation.

Annotated reviews are not available for download in order to protect the identity of reviewers who chose to remain anonymous.

---

## Round 0.2 · Minor Revisions

Please consider only a few mistakes pointed in the Figure #2.

Reviewer 1 ·

Basic reporting

no comment

Experimental design

no comment

Validity of the findings

no comment

Additional comments

I recommend acceptance of the revised and improved manuscript. One tiny mistake still remained in Figure 2. Now the size of the vertical lines marking the genera is correct, but their places should be exchanged. If the Authors check the branches, they will notice that the aviadenoviruses are the Meyer’s parrot AdV-1 and Fowl AdV-5 (just below the sturgeon AdV’s branch). The genus Siadenovirus is represented by 4 viruses, namely those from the tortoise, frog, plum-headed parakeet and the great tit. These 4 viruses form the branch of siadenoviruses. I am sorry for not being clear enough in my original review.

Reviewer 2 ·

Basic reporting

No comment.

Experimental design

No comment.

Validity of the findings

No comment.

Additional comments

No comment.

---

## Round 0.3 · accepted · Accept

The manuscript is accepted due the novelty of the findings, it contributes for the knowledge about adenoviruses in reptiles.

#